# Genome-wide identification and characterization of *miR396* family members and their target genes *GRF* in sorghum (*Sorghum bicolor* (L.) moench)

**Huiyan Wang[1,2], Yizhong Zhang[1,2]\*, Du Liang[1,2], Xiaojuan Zhang[1,2], Xinqi Fan[1,2], Qi Guo[1,2], Linfang Wang[1,2], Jingxue Wang[3], Qingshan Liu[2]\***

**1** Shanxi Key Laboratory of Sorghum Genetic and Germplasm Innovation, Sorghum Research Institute, Shanxi Agricultural University, Yuci, Shanxi Province, China, **2** Shanxi Key Laboratory of Minor Crops Germplasm Innovation and Molecular Breeding, State Key Laboratory of Sustainable Dryland Agriculture, Shanxi Agricultural University, Taiyuan, Shanxi Province, China, **3** School of Life Science, Shanxi University, Taiyuan, Shanxi Province, China

\* zhyzh225@163.com (YZ); 673673@126.com (QL)

**Data Availability Statement:** All relevant data are within the paper.

## Abstract

MicroRNAs (miRNAs) widely participate in plant growth and development. The *miR396* family, one of the most conserved miRNA families, remains poorly understood in sorghum. To reveal the evolution and expression pattern of *Sbi-miR396* gene family in sorghum, bioinformatics analysis and target gene prediction were performed on the sequences of the *Sbi-miR396* gene family members. The results showed that five *Sbi-miR396* members, located on chromosomes 4, 6, and 10, were identified at the whole-genome level. The secondary structure analysis showed that the precursor sequences of all five *Sbi-miR396* potentially form a stable secondary stem–loop structure, and the mature miRNA sequences were generated on the 5′ arm of the precursors. Sequence analysis identified the mature sequences of the five *sbi-miR396* genes were high identity, with differences only at the 1st, 9th and 21st bases at the 5' end. Phylogenetic analysis revealed that *Sbi-miR396a*, *Sbi-miR396b*, and *Sbi-miR396c* were clustered into Group I, and *Sbi-miR396d* and *Sbi-miR396e* were clustered into Group II, and all five *sbi-miR396* genes were closely related to those of maize and foxtail millet. Expression analysis of different tissue found that *Sbi-miR396d/e* and *Sbi-miR396a/b/c* were preferentially and barely expressed, respectively, in leaves, flowers, and panicles. Target gene prediction indicates that the growth-regulating factor family members (*SbiGRF1/2/3/4/5/6/7/8/10*) were target genes of *Sbi-miR396d/e*. Thus, *Sbi-miR396d/e* may affect the growth and development of sorghum by targeting *SbiGRF*s. In addition, expression analysis of different tissues and developmental stages found that all *Sbi-miR396* target genes, *SbiGRF*s, were barely expressed in leaves, root and shoot, but were predominantly expressed in inflorescence and seed development stage, especially *SbiGRF1/5/8*. Therefore, inhibition the expression of *sbi-miR396d/e* may increase the expression of *SbiGRF1/5/8*, thereby affecting floral organ and seed development in sorghum. These findings provide the basis for studying the expression of the *Sbi-mir396* family members and the function of their target genes.

**Funding:** This work was supported by the Special Project of Molecular Breeding Platform of Minor Crops of Shanxi Academy of Agricultural Sciences (YGC2019FZ5), the Shanxi Province Excellent Doctoral Work Award-Scientific Research Project (SXBYKY2022071), the Shanxi Province Brewing Sorghum Seed Industry Innovation and Superior Varieties Joint Research (2022), the Shanxi Province Modern Agricultural Minor Crops Industry Technology System (2022-03), and the Shanxi Provincial Key R & D Technology (International Cooperation) Project Grant (201803D421016). The funders had no role in study design, data collection and analysis, decision to publish, or preparation of the manuscript.

**Competing interests:** The authors have declared that no competing interests exist.

## Introduction

Sorghum (*Sorghum bicolor* [L.] Moench) is one of the world's top five grain crops, along with maize, rice, wheat, and barley. It is a staple food for more than 500 million people worldwide [1–3]. Owing to its excellent adaptability to heat and drought stress, sorghum is a key crop for safeguarding food security in response to climate change [4]. Recently, sorghum has also been widely used for different purposes, for example, as feed and forage, in brewing, and in biomass energy production, which has a significant effect on global agricultural production [5–8]. In recent years, with the continuous development of animal husbandry and brewery industries, sorghum production has struggled to meet the market demand [9, 10]. Crop yield is influenced by various factors, of which plant architecture has a significant effect [11, 12]. An ideal plant architecture is beneficial for a crop plant to develop an optimal shape for light exposure, which enables it to fully capture and use solar energy throughout the growth period and consequently maximizes the economic yield [11]. Plant architecture studies mainly focus on tillering and stem, leaf, panicle types, which are the traits determining crop yield [13, 14].

MicroRNAs (miRNAs) are a class of endogenous non-coding small RNAs commonly found in eukaryotic organisms, generally consisting of 20–22 nucleotides [15]. In plants, miRNA bind to the complementary sites within target mRNA to repress gene expression at the posttranscriptional level, thereby regulating plant growth, development, and response to adverse stress [16–18]. The *miR396* family, a conserved miRNA family in plants, has been identified in crops such as *Arabidopsis*, rice, wheat, tobacco, and soybean [19–23]. Most, but not all, growth-regulating factor genes (*GRF*s) are the target genes of *miR396* [24, 25]. *miR396* negatively regulates the expression of *GRF*s to participate in the regulation of leaf and floral organ morphogenesis, root and stem tissue elongation, and grain size and number, ultimately influencing plant architecture and crop yield. In *Arabidopsis*, overexpression of *miR396* decreases the mRNA levels of target *GRF*s, resulting in phenotypes such as short roots, small leaves, reduced number of petals and stamens, pistils with a single carpel, small siliques, and abnormal growing points at the tip of roots [25–27]. In rice, either plants overexpressing *miR396* or mutants carrying an insertionally inactivated *GRF* gene exhibit morphological changes such as retarded growth, reduced plant height, shortened roots, and narrowed leaves, whereas inhibition of *miR396* expression alters the panicle structure [28–30]. Furthermore, rice *miR396* (*OsmiR396d*) diminishes both gibberellin biosynthesis and signaling by inhibiting *OsGRF6* expression, which reduces plant height; it also relieves the inhibition of brassinolide signaling through suppression of *OsGRF4*, thereby positively regulating leaf angle [31]. The *OsmiR396–OsGRF3/4/8* module effectively regulates grain size and yield [14, 21, 32–34], and the *OsmiR396–OsGRF6/10* module regulates floral organ development. For example, rice plants overexpressing *OsmiR396* or *osgrf6/10* double mutant exhibit open husks, long sterile lemmas, and altered floral organ morphology, whereas rice plants with suppressed expression of *OsmiR396* or overexpression of *OsGRF6* exhibit a considerable increase in the number of secondary branches per panicle and glumes [35].

In addition to their essential roles in plant growth and development, *miR396* and its target *GRF* genes also enhance plant responses required to tolerate adverse stress conditions. Overexpression of *miR396* increases drought tolerance in crop plants. A previous study reports that *miR396a/b*-overexpressing *Arabidopsis* plants displayed a narrow leaf phenotype, with lower stomatal density and decreased transpiration rate; their survival rate was remarkably higher than that of empty vector controls after 14 d of drought treatment [25]. In tobacco lines overexpressing *miR396a-5p*, expression of *NtGRFs* is inhibited, osmoregulation and drought tolerance are enhanced, and reactive oxygen species levels are lowered [36]. *miR396* mutation may increase crop tolerance to saline–alkali and nitrogen deficiency stress [37–40]. Under nitrogen-deficient conditions, *osmir396ef* double mutants exhibit higher grain yield and

aboveground biomass. Recent research has also found that *miR396–GRF* markedly increases the regeneration efficiency of explants [41, 42].

The *miR396*-mediated *GRF* regulatory module (*miR396–GRF*) has shown potential application value in improving plant type, crop yield, and stress resistance and increasing the efficiency of plant genetic transformation. However, to date, members of the sorghum *miR396* (*Sbi-miR396*) family have not been identified and functionally analyzed. In the present study, bioinformatics methods were used to analyze the chromosomal distribution, phylogenetic relationships, base conservation, secondary structure, and target gene expression of the *Sbi-miR396* family. The findings provide molecular evidence for the evolution and biological functions of the *miR396* family in sorghum.

## Materials and methods

### Genome-wide identification and sequence analysis of *Sbi-miR396* and its target genes

The precursor sequences, mature sequences, and chromosomal localization information of the *Sbi-miR396* family members were downloaded from the miRBase database (https://www.mirbase.org/ftp.shtml). Base conservation of these sequences were analyzed using WebLogo 3 (http://weblogo.threeplusone.com/create.cgi) [40]. The sorghum whole-genome gff file (*Sorghum_bicolor*_NCBIv3.54.gff3) was downloaded from EnsemblPlants (http://plants.ensembl.org/Sorghum_bicolor/Info/Index). Physical position of each *Sbi-miR396* family gene was extracted from this file, and chromosomal localization map was constructed using Gene Location Visualize from GTF/GFF in TBtools software [43]. The secondary stem–loop structure of the *Sbi-miR396* precursors was predicted using RNAfold (http://rna.tbi.univie.ac.at./cgi-bin/RNAfold.cgi). A folding algorithm with basic options was employed to select the minimum free energy and partition function.

The target genes of *Sbi-miR396* were predicted using psRNATarget (http://plantgrn.noble.org/psRNATarget/) with default parameter values. Genes with the maximum expectation value ≤1.5 were selected as target genes, and the sites of interaction between *Sbi-miR396* and candidate target genes were analyzed. In addition, we estimated the molecular weight (MW) and isoelectric point (pI) of the candidate target proteins using ExPASy (https://web.expasy.org/protparam/), the functional annotation was performed using the National Center for Biotechnology Information (NCBI) database (http://www.ncbi.nlm.nih.gov/), and multiple alignment of the amino acid sequences were performed using DNAMAN 6.0 software.

### Phylogenetic analysis of *Sbi-miR396* and its target gene *SbiGRF*s

The *miR396* precursor sequences of maize (*Zea mays* L.), foxtail millet (*Setaria italica* [L.] P. Beauv), and rice (*Oryza sativa* L.) were downloaded from the miRBase database. The GRF protein sequences of maize, foxtail millet, rice, and *Arabidopsis* were downloaded from PlantTFDB database (http://planttfdb.gao-lab.org/index.php?sp=Sbi). Multiple alignment of *miR396* precursor sequences and GRF protein sequences were performed using ClustalW with default parameters. Phylogenetic trees were then constructed using the neighbor-joining method of MEGA v7.0 software with 1000 bootstraps and other default parameters.

### Sequence and structural analysis of *SbiGRF* gene family targeted by *Sbi-miR396*

Conserved motifs of SbiGRFs were analyzed using the multiple expectation maximization for motif elicitation (MEME, http://meme-suite.org/). Gene structure information was extracted

from the sorghum whole-genome gff file. These identified motifs and gene structures were further visualized by TBtools.

## Expression analysis of *Sbi-miR396* and its target gene *SbiGRF*s

Published RNA-seq data from different tissues or developmental stages of sorghum were downloaded from the PmiREN database (https://www.pmiren.com/browsehref4?wzid=209&mbt=5&type=sRNA-Seq#zqnav) and the Expression Atlas database (https://www.ebi.ac.uk/gxa/home) for the expression analysis of *Sbi-miR396* and its target gene *SbiGRF*s, respectively. The PmiREN database contained sorghum leaves, flowers, and panicles. The Expression Atlas database included different tissues and developmental stages, roots, shoots, leaves at seedling developmental stage; pistils and pollen at booting stage; inflorescences during the developmental process; and seeds during the developmental process. The expression level of Sbi-miR396 and SbiGRFs were estimated using the fragments per kilobase per million reads value (FPKM), and heatmaps were displayed by Tbtools.

## RNA isolation and qRT-PCR analysis

Three materials were used for qRT-PCR analysis, namely, 10480A, L17R and Jinnuo 3. Total RNA was extracted from young spikes of the inflorescence developmental stage and early development seeds using the RNAprep Pure Plant Kit (TIANGEN, Beijing, China) following the manufacturer's instructions. The purity and content of total RNA were detected by 1.0% agarose gel electrophoresis and NanoDrop 2000 (Thermo Fisher Scientific, USA). First-strand cDNA of miRNA and mRNA were synthesized with the Mir-X miRNA First-Strand Synthesis Kit and the PrimeScript$^{TM}$ RT reagent Kit with gDNA Eraser (Perfect Real Time) (Takara, Beijing, China), respectively. Quantitative real-time PCR (qRT-PCR) of *Sbi-miR396* and *SbiGRF*s were performed with three technical and three biological replicates in the LightCyclerTM 96 (Roche, Basel) using the Mir-X miRNA qRT-PCR TB Green Kit and the TB Green *Premix Ex Taq*$^{TM}$ II (Tli RNaseH Plus) (Takara, Beijing, China), respectively. The *U6* and *SbiGAPDH* gene were used as internal reference. The qRT-PCR reaction system of *Sbi-mi396* contained 10 μL 2×TB Green Advantage Premix, 0.4 μL (10 μM) RT-primer, 0.4 μL (10 μM) 5' primer (S1 Table), 2 μL cDNA template, 0.4 μL 50× Rox Reference Dye II, and 6.8 μL ddH$_2$O. The reaction procedure was as follows: denaturation at 95˚C for 10 s, followed by 40 cycles at 95˚C for 5 s and 60˚C for 20 s. The qRT-PCR reaction system of *SbiGRF*s contained 10 μL 2× TB Green *Premix Ex Taq*, 0.8 μL (10 μM) forward primer, 0.8 μL (10 μM) reverse primer (S1 Table), 2 μL cDNA template, 0.4 μL 50× Rox Reference Dye II, and 6.0 μL ddH$_2$O. The reaction procedure was as follows: denaturation at 95˚C for 3 min, followed by 40 cycles at 95˚C for 10 s and 60˚C for 30 s. After the reaction was completed, relative expression levels were calculated using the△CT method. Statistical significance was calculated using SPSS 19.0 software (SPSS Corp., Chicago, IL).

## Results

### Sequences and chromosomal localization of the *Sbi-miR396* family members

According to the miRBase database, the *miR396* family comprises five members in the sorghum genome, designated as *Sbi-miR396a*, *b*, *c*, *d*, and *e* (Table 1). The precursor sequences of the five *Sbi-miR396* members consisted of 88–186 bases. Bases at positions 1–22 at the 5′ end were highly conserved in the precursor sequences, whereas bases at other positions were less conserved (Fig 1A).

**Table 1. Basic information of *Sbi-miR396* family members.**

| miRNA | miRNA locus | Gene location | Mature miRNA | Base number | Star sequence | Base number |
|---|---|---|---|---|---|---|
| *Sbi-miR396a* | Sbi-MIR396a | Chr04: 66733975..66734099 (-) | **U**UCCACAGCUUUCUUGAACU**G** | 21 | G**U**UCAA**U**AAAG**CUGUG**GGA**AA** | 21 |
| *Sbi-miR396b* | Sbi-MIR396b | Chr10: 4466694..4466821 (+) | **U**UCCACAGCUUUCUUGAACU**G** | 21 | G**U**UCAA**U**AAAG**CUGUG**GGAA**A** | 21 |
| *Sbi-miR396c* | Sbi-MIR396c | Chr04: 66726748..66726909 (+) | **U**UCCACAGCUUUCUUGAACU**U** | 21 | G**G**UCAA**G**AAAG**CUGUG**GGAA**G** | 21 |
| *Sbi-miR396d* | Sbi-MIR396d | Chr04: 68279979..68280074 (-) | **C**UCCACAG**G**CUUUCUUGAACU**G** | 22 | G**U**UCAA**G**AAAG**UCCUU**GGAA**A** | 21 |
| *Sbi-miR396e* | Sbi-MIR396e | Chr06: 59881923..59882111 (+) | **U**UCCACAG**G**CUUUCUUGAACU**G** | 22 | G**U**UCAA**G**AAAG**CCCAU**GGAAA | 21 |

Mature *Sbi-miR396s* were all produced from the highly conserved bases at positions 1–22 of the precursor sequences. The mature sequences of *Sbi-miR396a*, *b*, and *c* each contained 21 bases, and *Sbi-miR396d* and *e* each contained 22 bases. All 21 bases were highly conserved between *Sbi-miR396a* and *Sbi-miR396b*. Excluding a dissimilar base at position 1 at the 5′ end, 21 bases were highly conserved between *Sbi-miR396d* and *Sbi-miR396e*. Excluding a dissimilar base at position 1 at the 3′ end, *Sbi-miR396c* were highly conserved with *Sbi-miR396a* and *Sbi-miR396b* for 20 bases. *Sbi-miR396d* and *Sbi-miR396e* contained an additional base, G, at position 9 from the 5′ end, compared with *Sbi-miR396a* and *Sbi-miR396b* (Fig 1B, Table 1). The star sequences of all five *Sbi-miR396* members comprised 21 bases. Of these, bases at positions 1, 3–6, 8–11, and 17–20 from the 5′ end were highly conserved; bases at positions 2, 12, 15, and 21 were moderately conserved; and bases at the remaining positions were poorly conserved (Fig 1C, Table 1).

The chromosomal localization results indicated that *Sbi-miR396a*, *Sbi-miR396c*, and *Sbi-miR396d* were all located on chromosome 4, whereas *Sbi-miR396b* and *Sbi-miR396e* were located on chromosomes 10 and 6, respectively (Fig 2, Table 1).

## Secondary structure of the *Sbi-miR396* precursor sequences

The precursor sequences of the five *Sbi-miR396* family members differed in length; *Sbi-miR396e* was the longest at 186 bp, and *Sbi-miR396a* was the shortest at 88 bp. The secondary structure prediction results showed that a stable secondary stem–loop structure could be formed in all precursor sequences. All mature miRNAs were located on the 5′ arm of the corresponding precursors and were highly conserved (Fig 3).

## Phylogenetic evolution of the *Sbi-miR396* family members

Phylogenetic analysis of the precursor sequences revealed that the *miR396* family members of sorghum, maize, foxtail millet, and rice were divided into three major groups (Groups I–III). *Sbi-miR396a*, *Sbi-miR396b*, and *Sbi-miR396c* clustered in Group I, whereas *Sbi-miR396d* and *Sbi-miR396e* clustered in Group II. Specifically, the *miR396* family members in sorghum were closely related to those in maize and foxtail millet, exhibiting clusters such as *Sbi-miR396a/Zma-miR396g/Sit-miR396c/Zma-miR396a* and *Sbi-miR396e/Sti-miR396b/Zma-miR396c*, but distantly related to those in rice (Fig 4).

## Tissue-specific expression of *Sbi-miR396* family members

Tissue-specific expression analysis revealed that *Sbi-miR396a* and *Sbi-miR396b* were barely expressed in young leaves, flowers, and young panicles of sorghum. *Sbi-miR396c* was slightly expressed in young leaves but not expressed in flowers and young panicles. In contrast, *Sbi-*

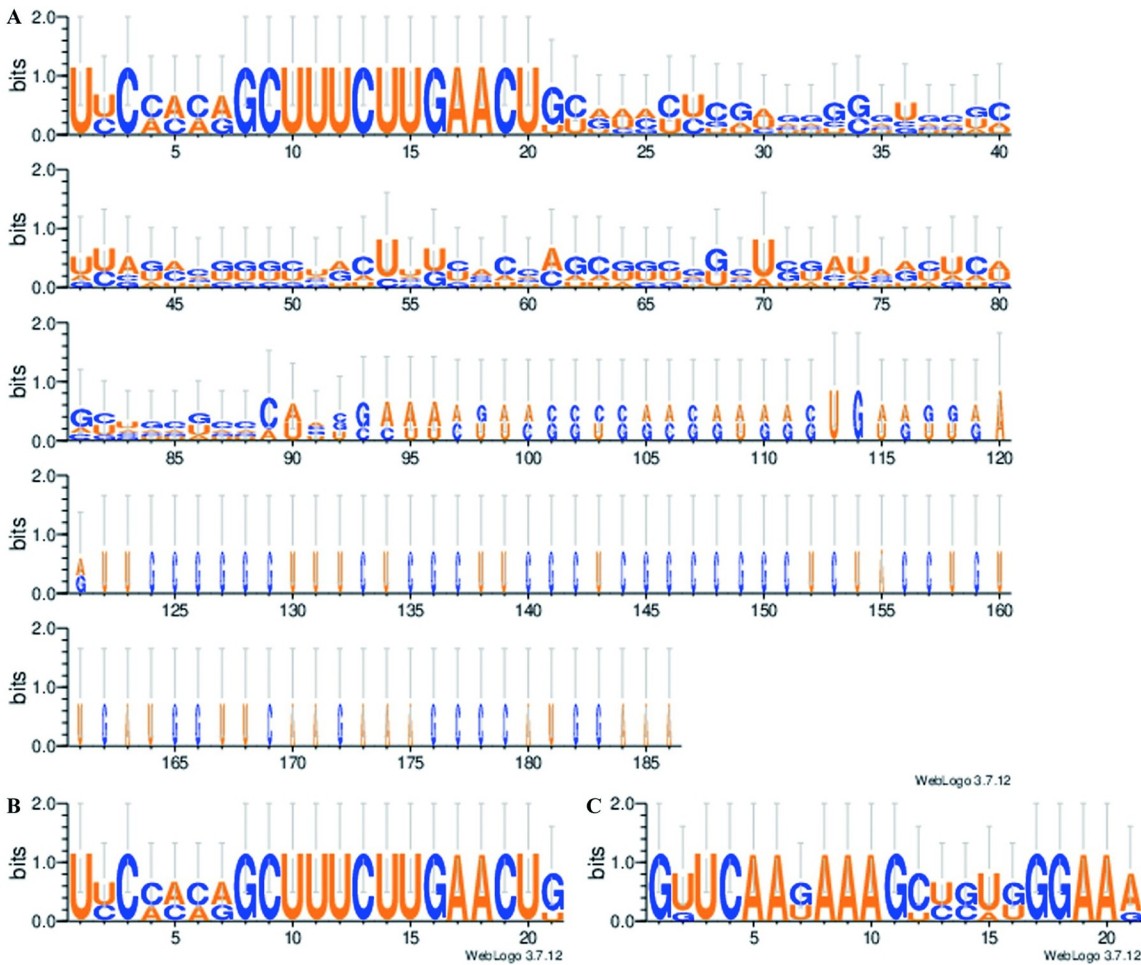

**Fig 1. Base conservation of *Sbi-miR396* sequences.** (A) Precursor sequence; (B) Mature sequence; (C) Star sequence.

*miR396d* and *Sbi-miR396e* were preferentially expressed in young leaves, flowers, and young panicles; their highest expression levels were observed in young leaves, followed by those in the flowers (Fig 5).

## Prediction and physicochemical features of target genes of *Sbi-miR396*

Taking into account the minimal expression of *Sbi-miR396a/b/c* and preferential expression of *Sbi-miR396d/e* in various tissues of sorghum, subsequent analyses were performed only for *Sbi-miR396d* and *Sbi-miR396e*. Table 2 lists the predicted target genes of *Sbi-miR396*. *Sbi-miR396d* and *Sbi-miR396e* were found to target the same nine *SbiGRF* family members, *SbiGRF1*, *2*, *3*, *4*, *5*, *6*, *8*, *9*, and *10*. The target sites of the *SbiGRF*s were all located in the third or second exon. The protein sequences of the nine SbiGFRs ranged from 271 to 601 amino acids in length, with a theoretical isoelectric point of 4.73–10.03 and molecular weight of 28.31–62.48 kDa.

## Phylogeny and classification of *SbiGRF*s

The GRF families of maize, foxtail millet, rice, and *Arabidopsis* had 15, 10, 12, and 9 members, respectively. The protein sequences of these GRFs and the nine GRFs of sorghum were used to

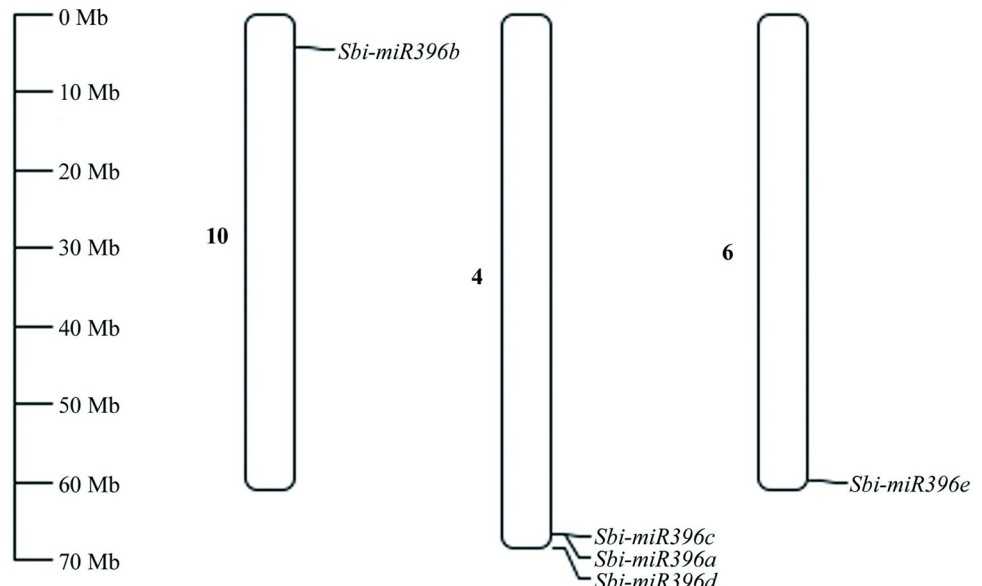

**Fig 2. Chromosome location of *Sbi-miR396* genes.**

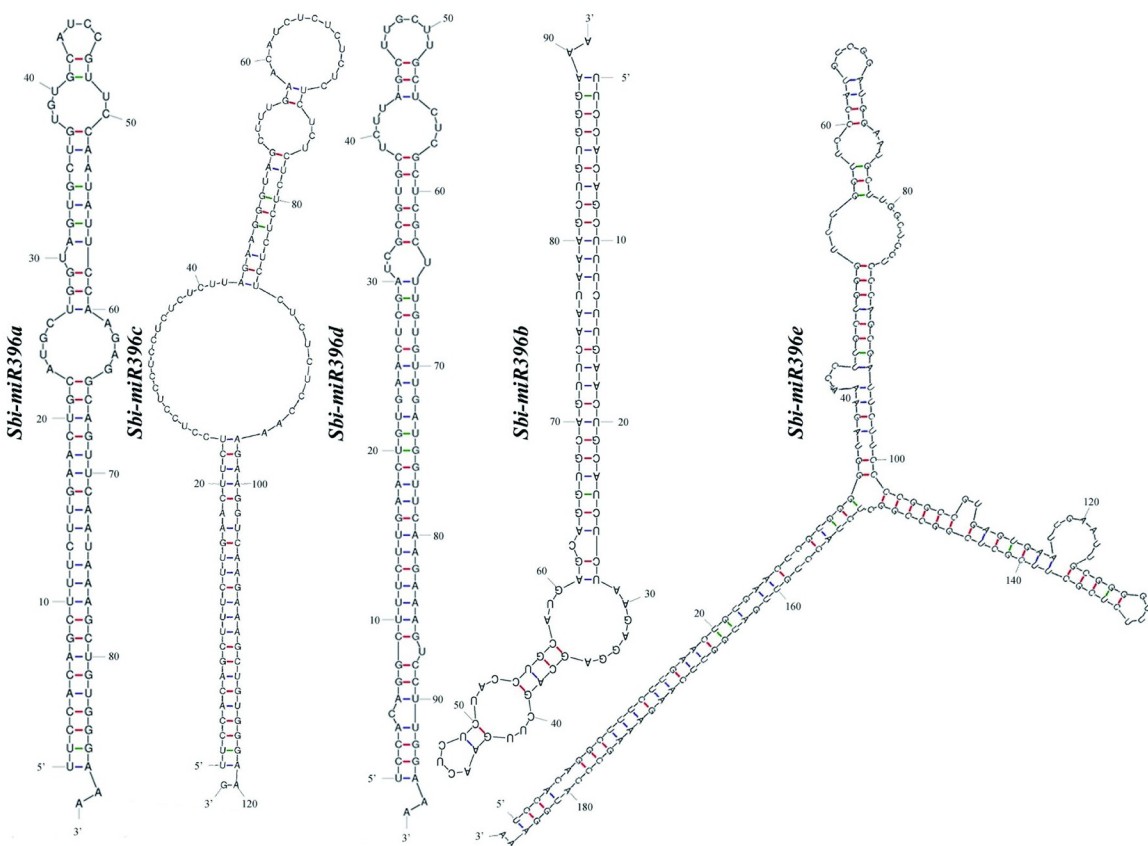

**Fig 3. Secondary structure of the precursor sequence of the *Sbi-miR396* family members.**

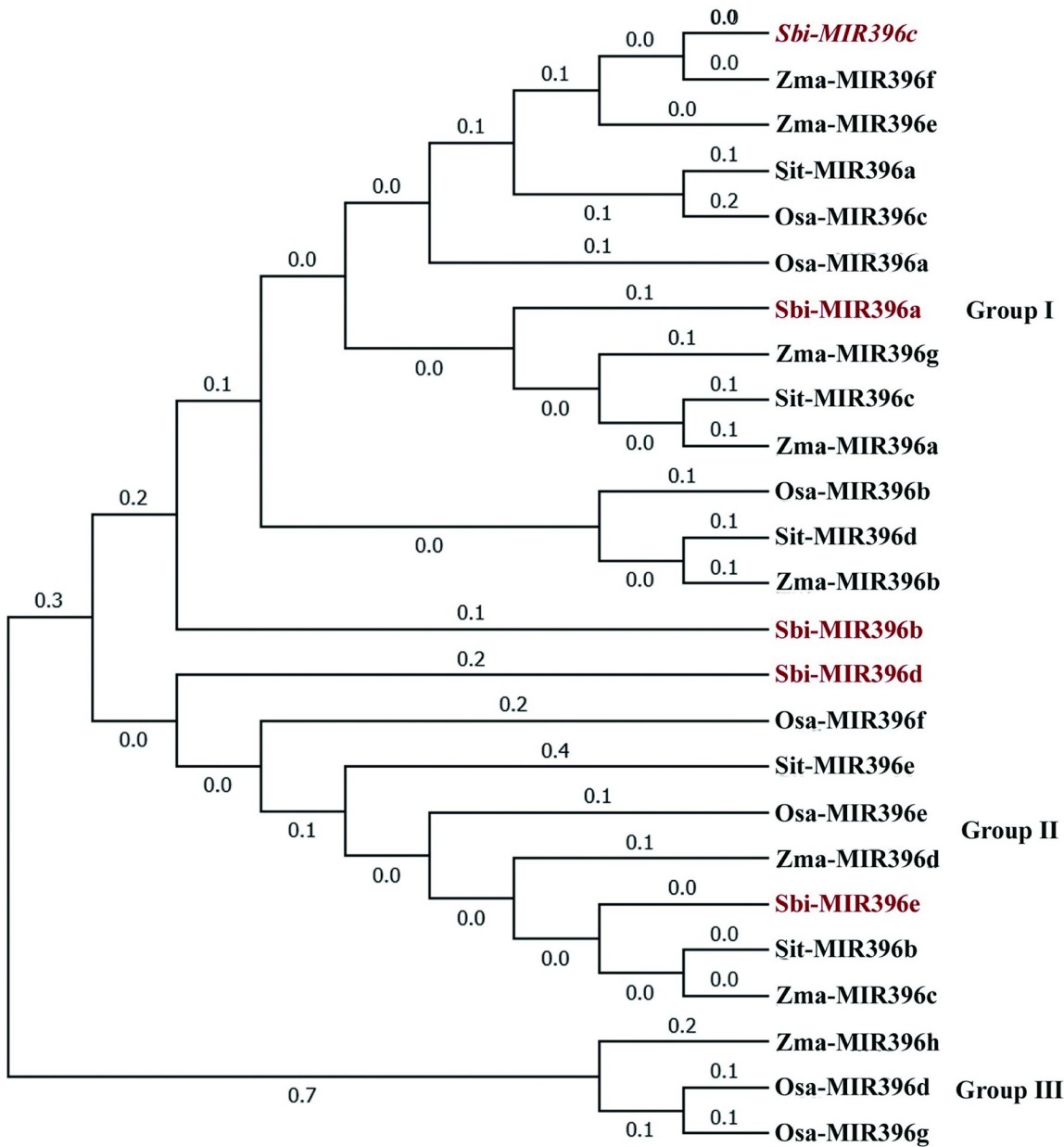

**Fig 4. Phylogenetic analysis of the *miR396* family.** Gene prefixtion *Sbi*, *Zma*, *Sit*, and *Osa* represent *Sorghum bicolor*, *Zea mays*, *Setaria italica*, and *Oryza sativa*, respectively.

construct a phylogenetic tree to analyze the evolutionary relationships among the GRF families. The 55 GRFs clustered into five groups (Group I–V; Fig 6). Sorghum, maize, foxtail millet, and rice GRFs were distributed in Group I, II, IV, and V. *Arabidopsis* GRFs were distributed in Group I, II, III, and IV. All GRF members in Group V were derived from dicotyledons. The results suggest that Group I, II, and IV already existed prior to differentiation of monocotyledonous and dicotyledonous plants, whereas Group V emerged after differentiation of monocotyledonous and dicotyledonous plants. The phylogenetic tree suggested that the GRF family of sorghum was most closely related to that of maize and foxtail millet, followed by rice, and most distantly related to that of the dicotyledon, *Arabidopsis*.

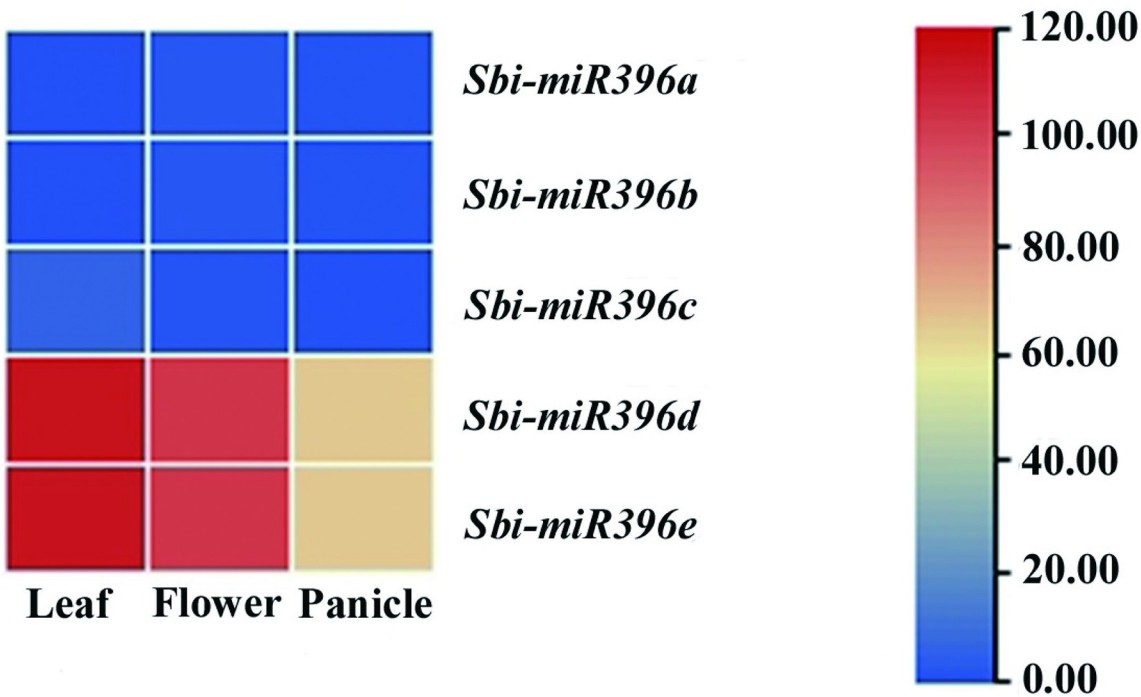

**Fig 5. Expression pattern of *Sbi-miR396* in different tissues.**

## Gene structure of *SbiGRF*s

The conserved protein sequences of the GRF family genes, putative targets of *Sbi-miR396*, were analyzed using MEME, which identified a total of five motifs. The N terminus of nine *SbiGFR* proteins contained intact conserved domains, WRC (Motif1) and QLQ (Motif2); in all cases, the QLQ domain was located in front of the WRC domain (Fig 7). In addition to the conserved amino acids Q/Gln-L/Leu-Q/GLn, the QLQ domain also comprised the

**Table 2. Prediction of *miR396d/e* target genes in *Sorghum bicolor*.**

| Target genes | Gene ID | Chromosomal position | Expected value | Number | Description of target gene | Target positions | pI | MV (kDa) | Length of amino acid |
|---|---|---|---|---|---|---|---|---|---|
| *SbiGRF1* | SORBI_3004G317000 | Chr04: 65267342..65268752 (-) | 0 | 2 | Growth-regulating factor 1 (*LOC8081935*) | Exon 2 | 8.36 | 42.89 | 396 |
| *SbiGRF2* | SORBI_3010G077200 | Chr10: 6375339..6376491 (+) | 0 | 2 | Growth-regulating factor 2 (*LOC8070748*) | Exon 2 | 9.73 | 35.94 | 338 |
| *SbiGRF3* | SORBI_3006G203400 | Chr06: 55407750..55411073 (-) | 0 | 1 | Growth-regulating factor 3 (*LOC8072276*) | Exon 3 | 8.57 | 41.03 | 378 |
| *SbiGRF4* | SORBI_3004G269900 | Chr04: 61417207..61421476 (+) | 0 | 3 | Growth-regulating factor 4 (*LOC8073168*) | Exon 3 | 7.53 | 43.09 | 415 |
| *SbiGRF5* | SORBI_3010G013500 | Chr10: 1131022..1133995 (-) | 0 | 1 | Growth-regulating factor 5 (*LOC8066917*) | Exon 3 | 8.33 | 53.06 | 497 |
| *SbiGRF6* | SORBI_3001G104500 | Chr01: 8002342..8005761 (-) | 1.5 | 1 | Growth-regulating factor 6 (*LOC8060535*) | Exon 3 | 6.98 | 62.48 | 601 |
| *SbiGRF8* | SORBI_3005G150900 | Chr05: 61985191..61987852 (+) | 1.5 | 1 | Growth-regulating factor 8 (*LOC110435841*) | Exon 3 | 4.73 | 28.31 | 271 |
| *SbiGRF9* | SORBI_3001G139800 | Chr01: 11114714..11117134 (-) | 0 | 1 | Growth-regulating factor 9 (*LOC8054183*) | Exon 3 | 8.59 | 40.01 | 371 |
| *SbiGRF10* | SORBI_3004G282601 | Chr04: 62447303..62448614 (-) | 0 | 1 | Growth-regulating factor 10 (*LOC8076036*) | Exon 2 | 9.29 | 47.57 | 435 |

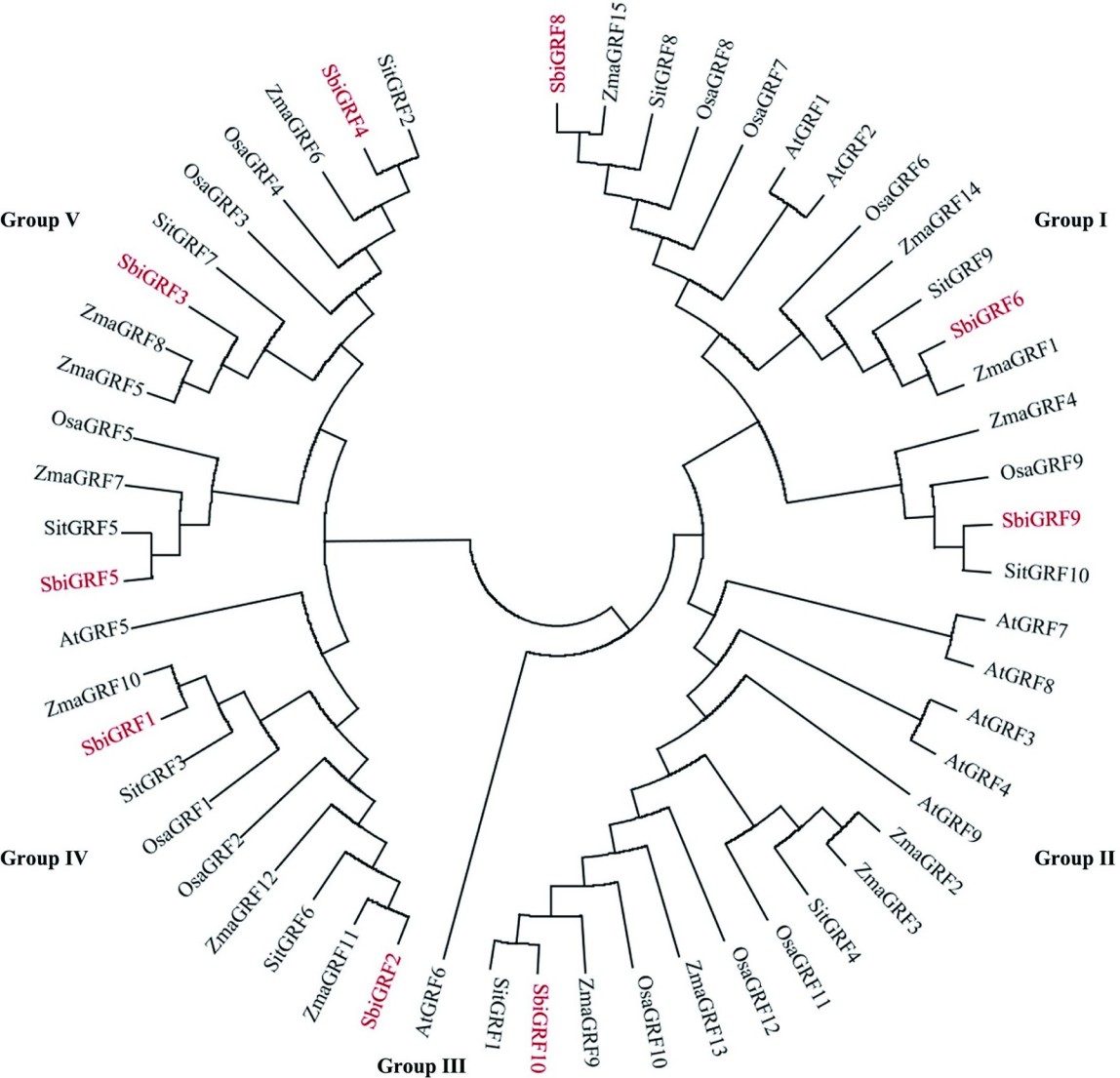

**Fig 6. Phylogenetic analysis of the GRFs family.** Gene prefixtion Sbi, Zma, Sit, Osa, and At represent *Sorghum bicolor*, *Zea mays*, *Setaria italica*, *Oryza sativa*, and *Arabidopsis thaliana*, respectively.

hydrophobic, acidic amino acid residues F/Phe, Y/Tyr, L/Leu, E/Glu, and P/Pro. The WRC domain contained the conserved amino acid residues W/Trp-R/Arg-C/Cys and the C3H motif consisting of three C/Cys and one H/His, immediately followed by the conserved RSRK-VE region (Fig 8). Furthermore, SbiGRF3–5 contained Motif3, 4, and 5; SbiGRF1 and 2 contained Motif3 and 4; SbiGRF6 and 8 contained Motif4; and SbiGRF10 contained Motif3.

Gene structure analysis revealed that the nine *SbiGRF* genes differed in the number of exons. *SbiGRF2* and 10 contained two exons, *SbiGRF1* and *5* contained three exons each, *SbiGRF4* contained five exons, whereas the remaining four *SbiGRF* genes contained four exons each (Fig 7).

## Expression analysis of *SbiGRF*s in different tissues and developmental stages

*miR396* affects plant growth and development by regulating the expression of its target genes *SbiGRFs*. We used online available transcriptome data of three tissues (leaf, shoot and root)

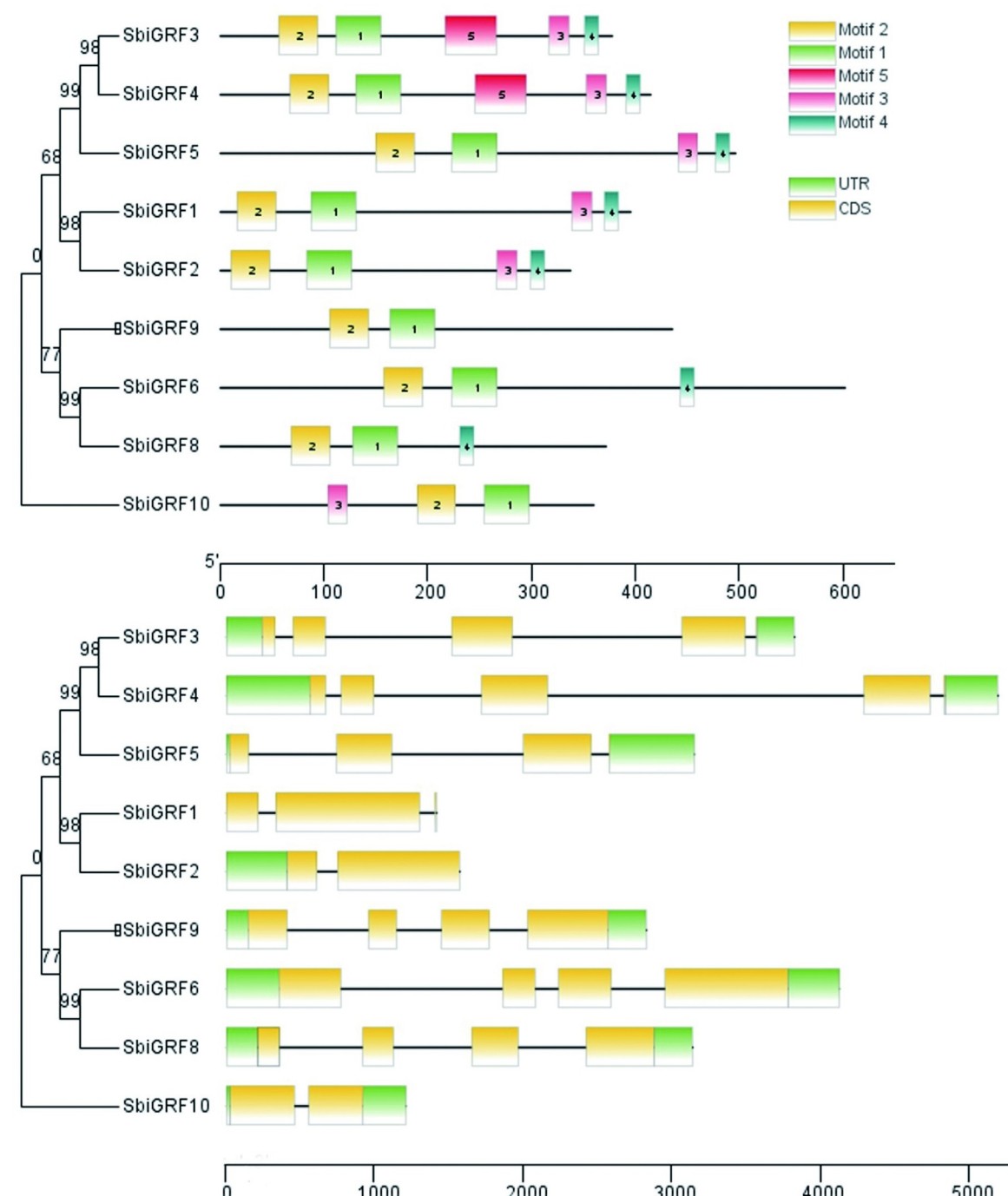

**Fig 7. Phylogenetic tree, motif distribution and gene structure of the *SbiGRF*s gene family.**

and eight developmental stages of inflorescence and seed (pistil and pollen at booting stage, Early inflorescence, Emerging inflorescence, Inflorescence size 1 to 5 millimeter, Inflorescence size 1 to 10 millimeter, Inflorescence size 1 to 2 centimeter, seed 5 days after pollination, seed 10 days after pollination) to investigate the expression patterns of *SbiGRF*s genes (Fig 9). It was found that *SbiGRF*s were predominantly expressed in inflorescence and seed. The expression patterns of *SbiGRF*s exhibited remarkable tissue specificity, indicating that they are mainly

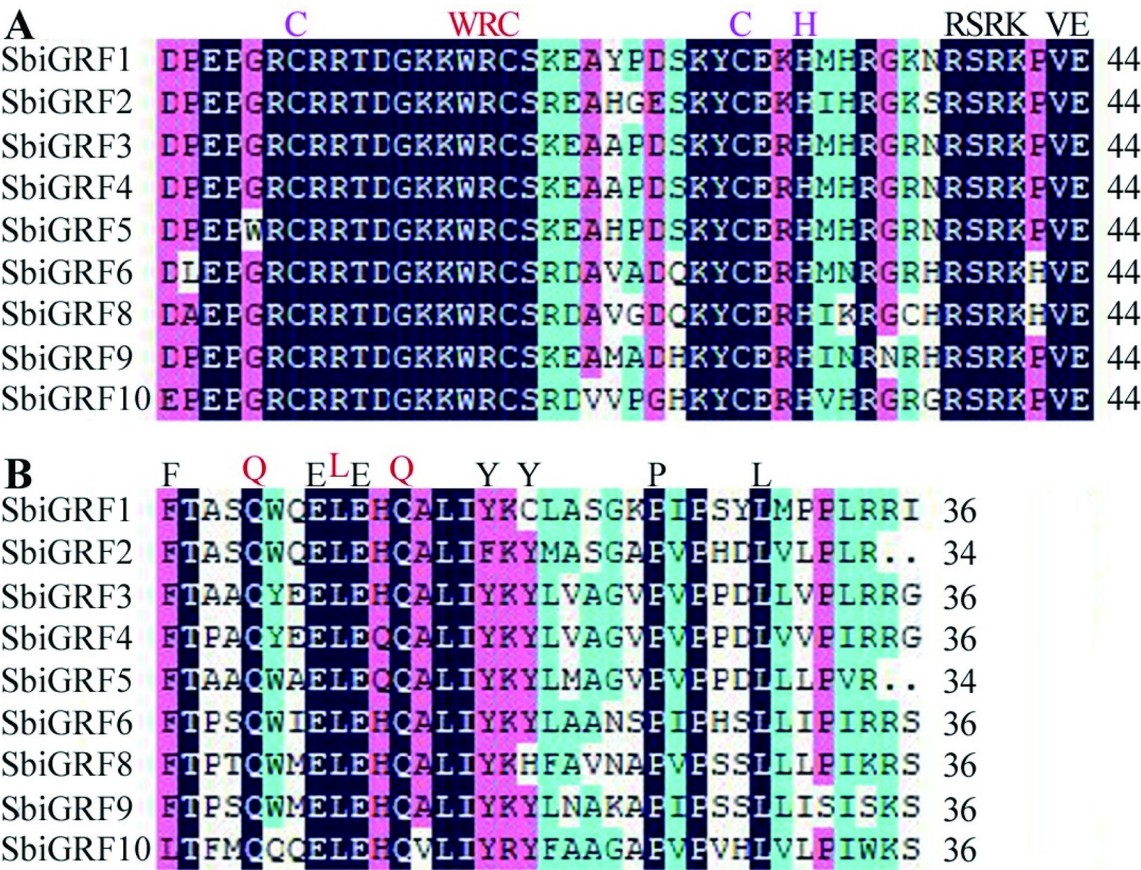

**Fig 8. Amino acid sequence alignment analysis of two conserved functional domains WRC and QLQ of *SbiGRF* protein family.** (A) WRC domain; (B) QLQ domain.

involved in regulating the growth and development of floral organs and seeds in sorghum. Specifically, at the seedling developmental stage, the expression levels of *SbiGRFs* were considerably low across in all tissues, especially in seedling leaves. At booting stage, *SbiGRFs* were barely expressed in pollen but preferentially expressed in pistils. During inflorescence development stage, the expression levels of *SbiGRFs* were higher after than before inflorescence emergence. Before inflorescence emergence (I1), *SbiGRF6/8* were expressed at slightly higher levels, whereas after inflorescence emergence(I3, I4 and I5), *SbiGRF5/8* were expressed at relatively higher levels, followed by *SbiGRF1/3/4/6*, and *SbiGRF2/10* showed low expression levels. During seed development stage, the expression levels of *SbiGRFs* in early developing seeds (P5S) were higher than those in intermediate developing seeds (P10S). In addition, *SbiGRF1/3/5/6/8* showed higher expression levels than other *SbiGRFs*.

To further clarify the role of *Sbi-miR396* and their target gene *SbiGRFs* in inflorescence and seed development, qRT-PCR was performed from young spikes of the inflorescence developmental stage and early development seeds in three sorghum varieties, and the results are shown in Fig 10. Consistent with the RNA-seq results, *Sbi-miR396a*, *Sbi-miR396b*, and *Sbi-miR396c* were expressed at low levels or not expressed in the young spikes of the inflorescence developmental stage and early development seeds, while *Sbi-miR396d* and *Sbi-miR396e* were preferentially expressed (Fig 10A). Moreover, *Sbi-miR396d* was expressed slightly higher than *Sbi-miR396e* at both developmental stages. Nine *Sbi-miR396* target genes, *SbiGRFs*, were expressed at both developmental stages, and most *SbiGRFs* were expressed slightly higher in

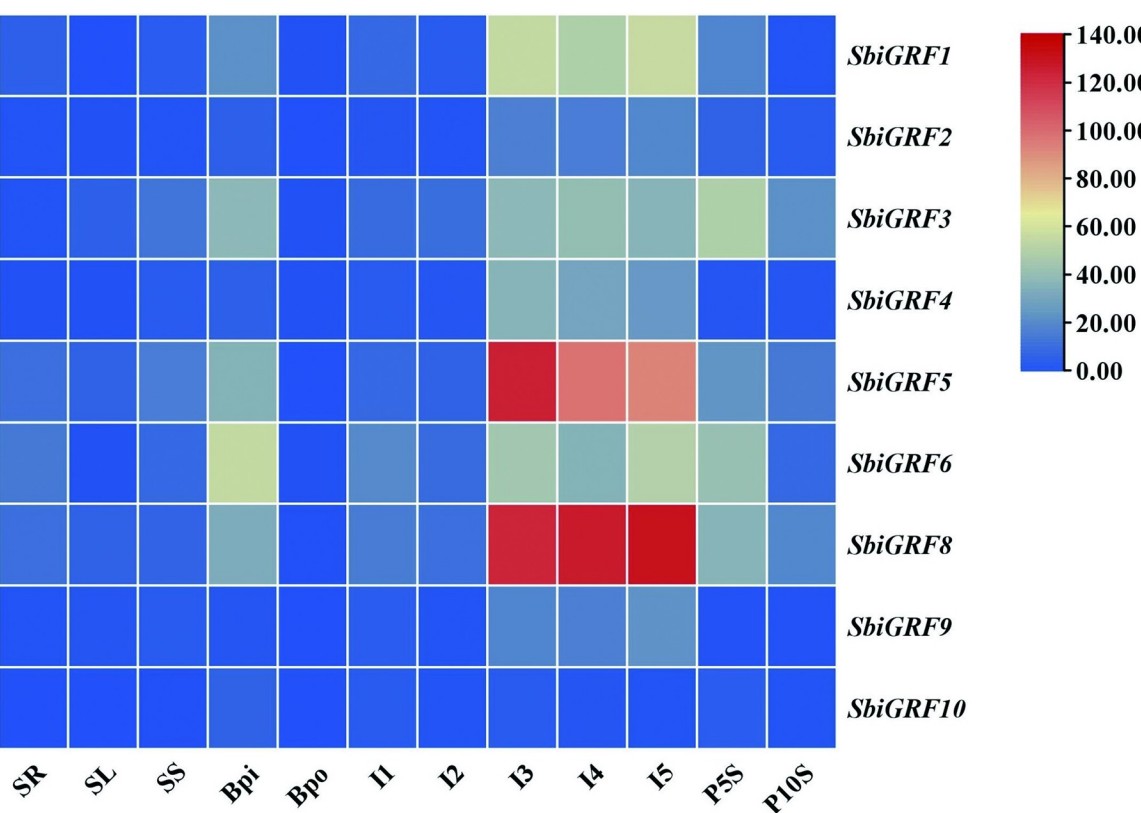

**Fig 9. Expression pattern of *SbiGRF*s in different tissues and developmental stages.** SR, SL, and SS represent root, leaf, and shoot at seedling developmental stage, respectively; Bpi and Bpo represent pistil and pollen at booting stage; I1, Early inflorescence; I2, Emerging inflorescence; I3, Inflorescence size 1 to 5 millimeter; I4, Inflorescence size 1 to 10 millimeter; I5, Inflorescence size 1 to 2 centimeter; P5S and P10S represent seed at 5 and 10 days after pollination.

the inflorescence development stage than in the seed development stage (Fig 10B). In sorghum varieties, *SbiGRF8* exhibited the highest expression level, followed by *SbiGRF1* and *SbiGRF5*. *SbiGRF9* and *SbiGRF4* showed the lowest expression level. *SbiGRF2*, *SbiGRF3*, *SbiGRF6*, and

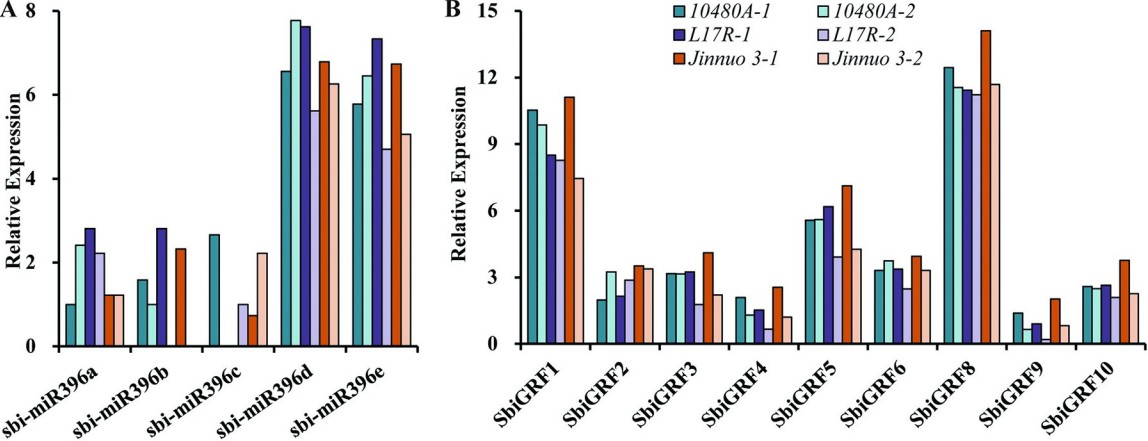

**Fig 10.** Expression analysis of *Sbi-miR396* (A) and *SbiGRF*s (B) in inflorescence and seed development. 1, young spikes of the inflorescence developmental stage; 2, early development seeds.

*SbiGRF10* showed intermediate expression level. Furthermore, *SbiGRF8* and *SbiGRF1* were expressed at about two times that of *SbiGRF5* and about ten times that of *SbiGRF9* and *SbiGRF4*. These findings suggest that *Sbi-miR396d/e* and *SbiGRF8/1/5* play a major role in inflorescence and seed development stage in sorghum.

## Discussion

Sorghum exhibits resistance to environmental stresses such as drought stress, flood stress, saline-alkali stress, nutrient deficiency, and heat stress. Consequently, it has comparative advantages when grown in fields with poor ecological and site conditions. However, compared with major crops such as rice and wheat or model crops such as Arabidopsis, sorghum has received less research attention regarding its molecular biology, especially miRNAs. To date, the miRBase database includes *miR396*s present in 52 species, with 221 mature sequences and 170 precursor sequences. Five plant species exhibit a high number of mature *miR396* sequences, namely soybean ($n$ = 15), rice ($n$ = 13), maize ($n$ = 13), *Brachypodium distachyon* ($n$ = 10), and *Aegilops tauschii* ($n$ = 10). Other plants such as wheat, sugarcane, and papaya have only one mature *miR396* sequence each. These findings indicate that the number of mature *miR396* sequences does not correspond exactly to the number of precursor sequences and some precursor sequences can produce *miR396* at both ends of the hairpin. Different mature sequences may originate from the same precursor sequence; in contrast, the same mature sequence may be derived from various precursor sequences. In the present study, it was found that the *Sbi-miR396* family comprises five mature and five precursor sequences in sorghum. Each mature *Sbi-miR396* is derived from the highly conserved positions 1–22 at the 5′ end of the corresponding precursor sequence. The five mature *Sbi-miR396* sequences are highly similar to with each other; except for three base differences between *Sbi-miR396c* and *Sbi-miR396d*, only one or no base difference was observed between any two *Sbi-miR396* sequences.

The *miR396* family shows high evolutionary conservation across species [44]. The present findings indicate that the *miR396* precursor sequences of sorghum have a close evolutionary relationship with those of foxtail millet and maize, suggesting that the evolutionary relationship of the *miR396* gene families is consistent with the phylogenetic relationship among the species themselves. The *Arabidopsis miR396* family includes four members, *Ath-miR396a-5p*, *Ath-miR396a-3p*, *Ath-miR396b-5p*, and *Ath-miR396b-3p*. The sequence alignment analysis demonstrated that the five *Sbi-miR396* sequences of sorghum have the same forms as the two mature sequences in *Arabidopsis*, *Ath-miR396a-5p* and *Ath-miR396b-5p*, indicating that the mature *miR396* sequences from the conserved region at the 5′ end are consistent between species, in agreement with the findings reported by Shan et al. [45]. Tissue-specific expression analysis revealed extremely low expression levels of *Sbi-miR396a*, *Sbi-miR396b*, and *Sbi-miR396c* in all tissues of sorghum, in contrast to the preferential expression of *Sbi-miR396d* and *Sbi-miR396e* in young leaves, flowers, and young panicles. The phylogenetic evolution analysis showed that *Sbi-miR396d* and *Sbi-miR396e* and *Osa-miR396e* and *Osa-miR396f* were clustered in the same group (Group II), so we speculated that they may have similar functions. Zhang et al. [39] reported that *Osa-miR396f* double mutant significantly increased panicle number, grain length, grain width, and yield compared with WT. In this study, qRT-PCR results showed *Sbi-miR396a*, *Sbi-miR396b*, and *Sbi-miR396c* were expressed at low levels or not expressed in the young spikes of the inflorescence developmental stage and early development seeds, while *Sbi-miR396d* and *Sbi-miR396e* were highly expressed at both developmental stages. These results suggest that only *Sbi-miR396d* and *Sbi-miR396e* are extensively involved in the growth and development of organs and tissues, especially the panicle development in

sorghum. Therefore, it is possible to change the panicle and grain development and increase the yield by regulating the expression level of *Sbi-miR396d* and *Sbi-miR396e* in sorghum.

GRFs are plant-specific transcription factors that play a key role in regulating various stages of plant growth and development. The N-terminus of GRF proteins commonly contains two conserved functional domains, QLQ and WRC [46]. In the present study, all nine GRFs targeted by *Sbi-miR396* also exhibited the intact conserved domains WRC (Motif1) and QLQ (Motif2), with QLQ located in front of WRC in all cases. The conserved RSRK-VE region present at the end of the conserved domain WRC corresponds to a highly conserved nucleotide coding sequence and is nearly complementary to the mature *Sbi-miR396* sequence. It has been reported that *miR396* cuts the mRNA sequence of the GRF gene between the sequences "CGUUCAAGAA" and "AGCNUGUGGAA" at this conserved site, thereby negatively regulating GRF gene expression [47]. The motif and gene structure analyses of *SbiGRF*s suggested that the nine *SbiGRF*s have different numbers of motifs and introns, implying that they perform differential functions in the regulation of the growth and development of sorghum.

In sorghum, *Sbi-miR396* influences the growth and development processes by targeting *SbiGRF*s and regulating their expression levels. Therefore, analysis of the expression patterns of the sorghum *GRF* genes facilitates the understanding of the function of the sorghum *miR396–GRF* module. In rice, most *GRF* genes are expressed in several tissues and participate in the regulation of multiple physiological processes during plant growth and development. However, expression analysis of *SbiGRF*s in different tissues and developmental stages revealed that they are mainly expressed in young inflorescences and early developing seeds, indicating remarkable tissue specificity. These results indicate that *SbiGRF*s mainly participate in the regulation of floral organ and seed growth and development in sorghum. The phylogenetic analysis demonstrated that *SbiGRF8/6/9* were clustered with *OsGRF6/7/8/9* in Group I, highlighting their close phylogenetic relationship. Research in rice has demonstrated that *OsGRF6* plays a vital role in regulating plant height, stem elongation, and floral organ development [31, 35]; *OsGRF7* mainly controls plant height, leaf length, and tiller number [13, 29]; *OsGRF8* regulates grain size and yield [33]; and *OsGRF9* is not involved in rice growth and development regulation but regulates rice blast resistance [48]. Regarding expression analysis of *SbiGRF*s in different tissues and developmental stages, the expression level of *SbiGRF8* was higher than that of other *SbiGRF*s in the young spikes of the inflorescence developmental stage and early development seeds. Thus, *SbiGRF8* may play a major role in regulating the development of floral organs and seeds in sorghum. *SbiGRF9* exhibited the lowest expression level at both developmental stages, suggesting that this gene is not associated with the regulation of sorghum growth and development. It may be functionally consistent with the homologous gene *OsGRF9* in rice and plays a role in disease resistance. *SbiGRF5/3/4* were clustered with *OsGRF3/4/5* in Group V. In rice, *OsGRF3/4* regulate grain size [21, 32, 34], and *OsGRF5* regulates plant height and inflorescence development [49]. In the present study, *SbiGRF5* were found to be highly expressed in inflorescence developmental and early development seeds of sorghum. In particular, *SbiGRF5* was preferentially expressed after inflorescences emergence, whereas *SbiGRF3/4* showed relatively low expression levels in various tissues. These results suggest that in sorghum, *SbiGRF5* was associated with the regulation of seed development and floral organ development. Furthermore, *SbiGRF2* and *SbiGRF10* also showed relatively low expression levels, whereas *SbiGRF1* exhibited very high expression levels at both developmental stages. These results imply that *SbiGRF*s may be both redundant and functionally specialized in regulating the growth and development of sorghum flower organs and seeds. Therefore, increasing the expression level of *SbiGRF*s, especially *SbiGRF8*, *SbiGRF1* and *SbiGRF5*, by means of genetic engineering may affect the panicle and grain development, and increase the yield in sorghum.

## Conclusions

In this study, a total of five *miR396* members were identified from sorghum genome. Notably, only *Sbi-miR396d* and *Sbi-miR396e* could regulate sorghum growth and development. *Sbi-miR396d/e* mainly target the growth-regulating factor genes *SbiGRF1/2/3/4/5/6/8/9/10*. However, only *SbiGRF1/5/8* were highly expressed in floral organ and early developing seeds. Thus, the *miR396d/e–GRF1/5/8* modules may be involved in regulation of sorghum floral organ and seed development, thus changing the sorghum spike structure and subsequently affecting its yield. Taken together, our study is the first comprehensive characterization of *miR396–GRF* in sorghum. All these findings provide the foundation to elucidate the molecular mechanism of *miR396–GRF* in sorghum floral organ and seed development.

## Supporting information

**S1 Table. Primers of qRT-PCR.**
(DOCX)

## Author Contributions

**Conceptualization:** Huiyan Wang, Yizhong Zhang, Qingshan Liu.

**Data curation:** Huiyan Wang, Du Liang, Xiaojuan Zhang, Xinqi Fan.

**Funding acquisition:** Qingshan Liu.

**Visualization:** Huiyan Wang, Yizhong Zhang, Qi Guo, Linfang Wang, Jingxue Wang.

**Writing – original draft:** Huiyan Wang.

**Writing – review & editing:** Huiyan Wang, Yizhong Zhang.

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
