## [Decision Letter · Decision Letter 0]

20 Feb 2023

PONE-D-23-02845Genome‑wide identification and characterization of miR396 family members and their target genes GRF in sorghum (Sorghum bicolor (L.) moench)PLOS ONE

Dear Dr. Zhang,

Thank you for submitting your manuscript to PLOS ONE. After careful consideration, we feel that it has merit but does not fully meet PLOS ONE’s publication criteria as it currently stands. Therefore, we invite you to submit a revised version of the manuscript that addresses the points raised during the review process.

ACADEMIC EDITOR:Dear Authors,

Two independent expert reviewers have finalized their revisions to the manuscript.

Based on their comments, the manuscript cannot be accepted in its present form but needs extensive revisions.

The main criticisms regarding the novelty of the work. They believe gene family analysis is routine work and many papers have been published in recent years. In this study, you should select some sorghum miR396 and GRF genes and perform qRT-PCR analysis to testify the results obtained from RNA-seq database..

The English language needs significant revision to meet the journal's standards.

I encourage the authors to proceed with revisions and submit the revised version of the manuscript.

We look forward to receiving your revised manuscript.

Kind regards,

Mojtaba Kordrostami, Ph.D.

Academic Editor

PLOS ONE

Journal Requirements:

2. Please amend your authorship list in your manuscript file to include author Dr. Jingxue Wang.

Reviewers' comments:

Reviewer's Responses to Questions

**Comments to the Author**

1. Is the manuscript technically sound, and do the data support the conclusions?

Reviewer #1: Yes

Reviewer #2: Partly

2. Has the statistical analysis been performed appropriately and rigorously? 

Reviewer #1: Yes

Reviewer #2: Yes

3. Have the authors made all data underlying the findings in their manuscript fully available?

Reviewer #1: Yes

Reviewer #2: Yes

4. Is the manuscript presented in an intelligible fashion and written in standard English?

Reviewer #1: Yes

Reviewer #2: Yes

5. Review Comments to the Author

Reviewer #1: “Genome‑wide identification and characterization of miR396 family members and their target genes GRF in sorghum (Sorghum bicolor (L.) Moench)” is a massive piece of work. I thank the authors for this work. Some comments/suggestions are given below point-wise that will help in improving the text and better presentation of the manuscript.

1. The quality of the images is very low and difficult to read and understand

2. Why did you investigate the secondary structure of miRNAs? What useful information does it provide for the purpose of this manuscript?

3. In figure 6, the clusters I to V are not specified. Therefore, the results written in the Phylogeny and classification of SbiGRFs section cannot be understood well. Which part of the dendrogram is cluster V?

4. In my opinion, it would be better to study based on GRFs (not miRNAs), and the analysis of GRFs-targeted miRNAs should be only one part of the manuscript because the information of miRNAs is not very useful but analyses such as duplication and selection pressure, exon-intron structure, promoter analysis, transcript-targeted miRNA and simple sequence repeat markers prediction, codon usage bias analysis, and three-dimensional structure of GRFs provide more information to the reader and it is easier to use the information for the desired genetic changes.

Taking the above into consideration, I recommend this manuscript for publication after applying the changes.

Reviewer #2: I wish thank the authors for this interesting manuscript. However, I have some critical comments, please, find them below:

1- The abstract must be rewritten with more detailed information in terms of results

2- Authors are using RNA-seq data to describe tissue specific expression levels of these genes. In my opinion, authors should select some sorghum miR396 and GRF genes and perform qRT-PCR analysis to testify the results obtained from RNA-seq database.

3- In M&M section the RNA-seq methodology has be described clearly

4- How were the gene expression count tables normalized, and what tool did you use?

5- The quality of figures should be increased significantly.

6- For the evolution study it is necessary to add more species (algae, moss, fern, some outgroup).

6. PLOS authors have the option to publish the peer review history of their article (what does this mean?). If published, this will include your full peer review and any attached files.

Reviewer #1: No

Reviewer #2: No

---

## [Author Response · Author response to Decision Letter 0]

10 Apr 2023

Dear Editor and Reviewers, 

We are very grateful for your critical comments and suggestions. According to these comments and suggestions, we have made revisions on the manuscript. We accepted all the suggestions of editor and reviewers. Our point-by-point responses to the editor and reviewers’ comments are as follows.

Editor:

1-Please amend your authorship list in your manuscript file to include author Dr. Jingxue Wang.

Answer: The author Dr. Jingxue Wang was added as follows:

Huiyan Wang1,2, Yizhong Zhang1,2*, Du Liang1,2, Xiaojuan Zhang1,2, Xinqi Fan1,2, Qi Guo1,2, Linfang Wang1,2, Jingxue Wang3, Qingshan Liu2*

3 School of Life Science, Shanxi University, Taiyuan 030031, Shanxi Province, China

Reviewer 1: 

1-The quality of the images is very low and difficult to read and understand

Answer: All images had been modified.

2-Why did you investigate the secondary structure of miRNAs? What useful information does it provide for the purpose of this manuscript?

Answer: The stable secondary stem–loop structure and its conservation are the judgment criteria for miRNA generation. Thus, the secondary structure of miRNAs must be investigated.

3-In figure 6, the clusters I to V are not specified. Therefore, the results written in the Phylogeny and classification of SbiGRFs section cannot be understood well. Which part of the dendrogram is cluster V?

Answer: The clusters I to V had been specified. The figure is as follows:

4-In my opinion, it would be better to study based on GRFs (not miRNAs), and the analysis of GRFs-targeted miRNAs should be only one part of the manuscript because the information of miRNAs is not very useful but analyses such as duplication and selection pressure, exon-intron structure, promoter analysis, transcript-targeted miRNA and simple sequence repeat markers prediction, codon usage bias analysis, and three-dimensional structure of GRFs provide more information to the reader and it is easier to use the information for the desired genetic changes.

Answer: In plants, miRNA bind to the complementary sites within target mRNA to repress gene expression at the posttranscriptional level, thereby regulating plant growth, development, and response to adverse stress. Currently, studies of plant miRNA mainly focus on a few model plants such as Arabidopsis and rice, which are less studied in coarse cereals. This study aimed to investigate how sorghum miRNA396 is involved in regulating its growth and development. Therefore, all studies were based on miRNA396, including its target gene GRF.

Reviewer 2:

1- The abstract must be rewritten with more detailed information in terms of results

Answer: We have rewritten the abstract. See the section “Abstract” at line 23-46 of page 1:

MicroRNAs (miRNAs) widely participate in plant growth and development. The miR396 family, one of the most conserved miRNA families, remains poorly understood in sorghum. To reveal the evolution and expression pattern of Sbi-miR396 gene family in sorghum, bioinformatics analysis and target gene prediction were performed on the sequences of the Sbi-miR396 gene family members. The results showed that five Sbi-miR396 members, located on chromosomes 4, 6, and 10, were identified at the whole-genome level. The secondary structure analysis showed that the precursor sequences of all five Sbi-miR396 potentially form a stable secondary stem–loop structure, and the mature miRNA sequences were generated on the 5′ arm of the precursors. Sequence analysis identified the mature sequences of the five sbi-miR396 genes were high identity, with differences only at the 1st, 9th and 21st bases at the 5' end. Phylogenetic analysis revealed that Sbi-miR396a, Sbi-miR396b, and Sbi-miR396c were clustered into Group I, and Sbi-miR396d and Sbi-miR396e were clustered into Group II, and all five sbi-miR396 genes were closely related to those of maize and foxtail millet. Expression analysis of different tissue found that Sbi-miR396d/e and Sbi-miR396a/b/c were preferentially and barely expressed, respectively, in leaves, flowers, and panicles. Target gene prediction indicates that the growth-regulating factor family members (SbiGRF1/2/3/4/5/6/7/8/10) were target genes of Sbi-miR396d/e. Thus, Sbi-miR396d/e may affect the growth and development of sorghum by targeting SbiGRFs. In addition, expression analysis of different tissues and developmental stages found that all Sbi-miR396 target genes, SbiGRFs, were barely expressed in leaves, root and shoot, but were predominantly expressed in inflorescence and seed development stage, especially SbiGRF1/5/8. Therefore, inhibition the expression of sbi-miR396d/e may increase the expression of SbiGRF1/5/8, thereby affecting floral organ and seed development in sorghum. These findings provide the basis for studying the expression of the Sbi-mir396 family members and the function of their target genes.

2- Authors are using RNA-seq data to describe tissue specific expression levels of these genes. In my opinion, authors should select some sorghum miR396 and GRF genes and perform qRT-PCR analysis to testify the results obtained from RNA-seq database.

Answer: It had been corrected. See the section “Expression analysis of SbiGRFs in different tissues and developmental stages” at line 316-325 of page 16:

To further clarify the role of Sbi-miR396 and their target gene SbiGRFs in inflorescence and seed development, qRT-PCR was performed from young spikes of the inflorescence developmental stage and early development seeds, and the results are shown in Fig 10. Consistent with the RNA-seq results, Sbi-miR396a, Sbi-miR396b, and Sbi-miR396c were barely expressed in the young spikes of the inflorescence developmental stage and early development seeds, while Sbi-miR396d and Sbi-miR396e were preferentially expressed (Fig 10A). All Sbi-miR396 target genes, SbiGRFs, were expressed at both developmental stages, and were expressed slightly higher in the inflorescence development stage than in the seed development stage. Furthermore, SbiGRF1/5/8 was predominantly expressed at both developmental stages, while other SbiGRFs showed slightly lower expression levels (Fig 10B).

3- In M&M section the RNA-seq methodology has be described clearly

Answer: It had been corrected. See the section “RNA isolation and qRT-PCR analysis” at line 155-177 of page 7-8:

RNA isolation and qRT-PCR analysis

Three materials were used for qRT-PCR analysis, namely, 10480A, L17R and Jinnuo 3. Total RNA was extracted from young spikes of the inflorescence developmental stage and early development seeds using the RNAprep Pure Plant Kit (TIANGEN, Beijing, China) following the manufacturer’s instructions. The purity and content of total RNA were detected by 1.0% agarose gel electrophoresis and NanoDrop 2000 (Thermo Fisher Scientific，USA). First-strand cDNA of miRNA and mRNA were synthesized with the Mir-X miRNA First-Strand Synthesis Kit and the PrimeScriptTM RT reagent Kit with gDNA Eraser (Perfect Real Time) (Takara, Beijing, China), respectively. Quantitative real-time PCR (qRT-PCR) of Sbi-miR396 and SbiGRFs were performed with three technical and three biological replicates in the LightCyclerTM 96 (Roche, Basel) using the Mir-X miRNA qRT-PCR TB Green Kit and the TB Green Premix Ex TaqTM II (Tli RNaseH Plus) (Takara, Beijing, China), respectively. The U6 and SbiGAPDH gene were used as internal reference. The qRT-PCR reaction system of Sbi-mi396 contained 10 μL 2×TB Green Advantage Premix, 0.4 μL (10 μM) RT-primer, 0.4 μL (10 μM) 5' primer (S1 Table), 2 μL cDNA template, 0.4 μL 50× Rox Reference Dye II, and 6.8 μL ddH2O. The reaction procedure was as follows: denaturation at 95 ℃ for 10 s, followed by 40 cycles at 95 ℃ for 5 s and 60 ℃ for 20 s. The qRT-PCR reaction system of SbiGRFs contained 10 μL 2× TB Green Premix Ex Taq, 0.8 μL (10 μM) forward primer, 0.8 μL (10 μM) reverse primer (S1 Table), 2 μL cDNA template, 0.4 μL 50× Rox Reference Dye II, and 6.0 μL ddH2O. The reaction procedure was as follows: denaturation at 95 ℃ for 3 min, followed by 40 cycles at 95 ℃ for 10 s and 60 ℃ for 30 s. After the reaction was completed, relative expression levels were calculated using the△CT method. Statistical significance was calculated using SPSS 19.0 software (SPSS Corp., Chicago, IL).

4- How were the gene expression count tables normalized, and what tool did you use?

Answer: The U6 and SbiGAPDH gene were used as the internal reference to normalize the gene expression data in qRT-PCR of Sbi-miR396 and SbiGRFs. This information is described in the section “RNA isolation and qRT-PCR analysis” at line 166-177 of page 8.

5- The quality of figures should be increased significantly.

Answer: All images have been modified.

6- For the evolution study it is necessary to add more species (algae, moss, fern, some outgroup).

Answer: Algae, moss, and fern do not have miR396.

---

## [Decision Letter · Decision Letter 1]

17 Apr 2023

PONE-D-23-02845R1Genome‑wide identification and characterization of miR396 family members and their target genes GRF in sorghum (Sorghum bicolor (L.) moench)PLOS ONE

Dear Dr. Zhang,

Thank you for submitting your manuscript to PLOS ONE. After careful consideration, we feel that it has merit but does not fully meet PLOS ONE’s publication criteria as it currently stands. Therefore, we invite you to submit a revised version of the manuscript that addresses the points raised during the review process. Please submit your revised manuscript by Jun 01 2023 11:59PM. If you will need more time than this to complete your revisions, please reply to this message or contact the journal office at plosone@plos.org. Please include the following items when submitting your revised manuscript:A rebuttal letter that responds to each point raised by the academic editor and reviewer(s). You should upload this letter as a separate file labeled 'Response to Reviewers'.A marked-up copy of your manuscript that highlights changes made to the original version. You should upload this as a separate file labeled 'Revised Manuscript with Track Changes'.An unmarked version of your revised paper without tracked changes. You should upload this as a separate file labeled 'Manuscript'.If applicable, we recommend that you deposit your laboratory protocols in protocols.io to enhance the reproducibility of your results. Protocols.io assigns your protocol its own identifier (DOI) so that it can be cited independently in the future. For instructions see: https://journals.plos.org/plosone/s/submission-guidelines#loc-laboratory-protocols. Additionally, PLOS ONE offers an option for publishing peer-reviewed Lab Protocol articles, which describe protocols hosted on protocols.io. Read more information on sharing protocols at https://plos.org/protocols?utm_medium=editorial-email&utm_source=authorletters&utm_campaign=protocols.

We look forward to receiving your revised manuscript.

Kind regards,

Mojtaba Kordrostami, Ph.D.

Academic Editor

PLOS ONE

Journal Requirements:

Additional Editor Comments:

Dear collegues

I have received reports from 2 experts in this field.

Please revise the manuscript before final acceptance.

Regards

Reviewers' comments:

Reviewer's Responses to Questions

**Comments to the Author**

1. If the authors have adequately addressed your comments raised in a previous round of review and you feel that this manuscript is now acceptable for publication, you may indicate that here to bypass the “Comments to the Author” section, enter your conflict of interest statement in the “Confidential to Editor” section, and submit your "Accept" recommendation.

Reviewer #1: All comments have been addressed

Reviewer #2: (No Response)

2. Is the manuscript technically sound, and do the data support the conclusions?

Reviewer #1: Yes

Reviewer #2: Yes

3. Has the statistical analysis been performed appropriately and rigorously? 

Reviewer #1: Yes

Reviewer #2: No

4. Have the authors made all data underlying the findings in their manuscript fully available?

Reviewer #1: Yes

Reviewer #2: Yes

5. Is the manuscript presented in an intelligible fashion and written in standard English?

Reviewer #1: Yes

Reviewer #2: Yes

6. Review Comments to the Author

Reviewer #1: I thank the authors of “Genome‑wide identification and characterization of miR396 family members and their target genes GRF in sorghum (Sorghum bicolor (L.) Moench)” for the revised manuscript. Based on the authors' responses to my comments, the revised article is ready for publication.

Reviewer #2: I would like to express my gratitude to the authors for incorporating my suggestions. Nevertheless, I have a few additional comments:

1- The statistical analysis conducted to examine the expression pattern of the selected genes lacked clarity, making it difficult to determine which gene expression changes were significant. Please revise the qRT-PCR results and Figure 10 accordingly.

2- Although qRT-PCR analysis was performed and its findings were presented, they were not discussed section of the article. Please address this in the manuscript.

7. PLOS authors have the option to publish the peer review history of their article (what does this mean?). If published, this will include your full peer review and any attached files.

Reviewer #1: No

Reviewer #2: No

---

## [Author Response · Author response to Decision Letter 1]

21 Apr 2023

Dear Editor,

We have made revisions on the manuscript following your comments and suggestions. All changes made in the manuscript are in red. Our point-by-point responses to the editor’ comments are as follows.

Reviewer 2: 

1- The statistical analysis conducted to examine the expression pattern of the selected genes lacked clarity, making it difficult to determine which gene expression changes were significant. Please revise the qRT-PCR results and Figure 10 accordingly.

Answer: qRT-PCR results and Figure 10 had been revised. See the qRT-PCR results at line 320-331 of page 16:

Sbi-miR396a, Sbi-miR396b, and Sbi-miR396c were expressed at low levels or not expressed in the young spikes of the inflorescence developmental stage and early development seeds, while Sbi-miR396d and Sbi-miR396e were preferentially expressed (Fig 10A). Moreover, Sbi-miR396d was expressed slightly higher than Sbi-miR396e at both developmental stages. Nine Sbi-miR396 target genes, SbiGRFs, were expressed at both developmental stages, and most SbiGRFs were expressed slightly higher in the inflorescence development stage than in the seed development stage (Fig 10B). In sorghum varieties, SbiGRF8 exhibited the highest expression level, followed by SbiGRF1 and SbiGRF5. SbiGRF9 and SbiGRF4 showed the lowest expression level. SbiGRF2, SbiGRF3, SbiGRF6, and SbiGRF10 showed intermediate expression level. Furthermore, SbiGRF8 and SbiGRF1 were expressed at about two times that of SbiGRF5 and about ten times that of SbiGRF9 and SbiGRF4. These findings suggest that Sbi-miR396d/e and SbiGRF8/1/5 play a major role in inflorescence and seed development stage in sorghum. 

2- Although qRT-PCR analysis was performed and its findings were presented, they were not discussed section of the article. Please address this in the manuscript.

Answer: It had been added. 

See the section “Discussion” at line 368-379 of page 18:

The phylogenetic evolution analysis showed that Sbi-miR396d and Sbi-miR396e and Osa-miR396e and Osa-miR396f were clustered in the same group (Group II), so we speculated that they may have similar functions. Zhang et al. [39] reported that Osa-miR396f double mutant significantly increased panicle number, grain length, grain width, and yield compared with WT. In this study, qRT-PCR results showed Sbi-miR396a, Sbi-miR396b, and Sbi-miR396c were expressed at low levels or not expressed in the young spikes of the inflorescence developmental stage and early development seeds, while Sbi-miR396d and Sbi-miR396e were highly expressed at both developmental stages. These results suggest that only Sbi-miR396d and Sbi-miR396e are extensively involved in the growth and development of organs and tissues, especially the panicle development in sorghum. Therefore, it is possible to change the panicle and grain development and increase the yield by regulating the expression level of Sbi-miR396d and Sbi-miR396e in sorghum.

See the section “Discussion” at line 400-425 of page 19-20:

The phylogenetic analysis demonstrated that SbiGRF8/6/9 were clustered with OsGRF6/7/8/9 in Group I, highlighting their close phylogenetic relationship. Research in rice has demonstrated that OsGRF6 plays a vital role in regulating plant height, stem elongation, and floral organ development[31, 35]; OsGRF7 mainly controls plant height, leaf length, and tiller number[13, 29]; OsGRF8 regulates grain size and yield[33]; and OsGRF9 is not involved in rice growth and development regulation but regulates rice blast resistance[48]. Regarding expression analysis of SbiGRFs in different tissues and developmental stages, the expression level of SbiGRF8 was higher than that of other SbiGRFs in the young spikes of the inflorescence developmental stage and early development seeds. Thus, SbiGRF8 may play a major role in regulating the development of floral organs and seeds in sorghum. SbiGRF9 exhibited the lowest expression level at both developmental stages, suggesting that this gene is not associated with the regulation of sorghum growth and development. It may be functionally consistent with the homologous gene OsGRF9 in rice and plays a role in disease resistance. SbiGRF5/3/4 were clustered with OsGRF3/4/5 in Group V. In rice, OsGRF3/4 regulate grain size[21, 32, 34], and OsGRF5 regulates plant height and inflorescence development [49]. In the present study, SbiGRF5 were found to be highly expressed in inflorescence developmental and early development seeds of sorghum. In particular, SbiGRF5 was preferentially expressed after inflorescences emergence, whereas SbiGRF3/4 showed relatively low expression levels in various tissues. These results suggest that in sorghum, SbiGRF5 was associated with the regulation of seed development and floral organ development. Furthermore, SbiGRF2 and SbiGRF10 also showed relatively low expression levels, whereas SbiGRF1 exhibited very high expression levels at both developmental stages. These results imply that SbiGRFs may be both redundant and functionally specialized in regulating the growth and development of sorghum flower organs and seeds. Therefore, increasing the expression level of SbiGRFs, especially SbiGRF8, SbiGRF1 and SbiGRF5, by means of genetic engineering may affect the panicle and grain development, and increase the yield in sorghum.

---

## [Editor Report · Decision Letter 2]

25 Apr 2023

Genome‑wide identification and characterization of miR396 family members and their target genes GRF in sorghum (Sorghum bicolor (L.) moench)

PONE-D-23-02845R2

Dear Dr. Zhang,

We’re pleased to inform you that your manuscript has been judged scientifically suitable for publication and will be formally accepted for publication once it meets all outstanding technical requirements.

Kind regards,

Mojtaba Kordrostami, Ph.D.

Academic Editor

PLOS ONE

Additional Editor Comments (optional):

The paper can be accepted now.
---

## [Editor Report · Acceptance letter]

2 May 2023

PONE-D-23-02845R2 

Genome‑wide identification and characterization of miR396 family members and their target genes *GRF* in sorghum (*Sorghum bicolor* (L.) moench) 

Dear Dr. Zhang:

I'm pleased to inform you that your manuscript has been deemed suitable for publication in PLOS ONE. Congratulations! Your manuscript is now with our production department. 

Kind regards, 

on behalf of

Dr. Mojtaba Kordrostami 

Academic Editor

PLOS ONE